# Applying a Chemogeographic Strategy for Natural Product Discovery from the Marine Cyanobacterium *Moorena bouillonii*

**DOI:** 10.3390/md18100515

**Published:** 2020-10-14

**Authors:** Christopher A. Leber, C. Benjamin Naman, Lena Keller, Jehad Almaliti, Eduardo J. E. Caro-Diaz, Evgenia Glukhov, Valsamma Joseph, T. P. Sajeevan, Andres Joshua Reyes, Jason S. Biggs, Te Li, Ye Yuan, Shan He, Xiaojun Yan, William H. Gerwick

**Affiliations:** 1Center for Marine Biotechnology and Biomedicine, Scripps Institution of Oceanography, University of California San Diego, La Jolla, CA 92093, USA; cleber@ucsd.edu (C.A.L.); bnaman@nbu.edu.cn (C.B.N.); le.keller85@gmail.com (L.K.); jalmaliti@ucsd.edu (J.A.); eduardo.caro1@upr.edu (E.J.E.C.-D.); eglukhov@ucsd.edu (E.G.); valsamma@cusat.ac.in (V.J.); sajeev@cusat.ac.in (T.P.S.); 2Li Dak Sum Yip Yio Chin Kenneth Li Marine Biopharmaceutical Research Center, Department of Marine Pharmacy, College of Food and Pharmaceutical Sciences, Ningbo University, Ningbo 315800, China; telinbu@163.com (T.L.); 23yuanye@163.com (Y.Y.); heshan@nbu.edu.cn (S.H.); yanxiaojun@nbu.edu.cn (X.Y.); 3Department Microbial Natural Products, Helmholtz-Institute for Pharmaceutical Research Saarland (HIPS), Helmholtz Centre for Infection Research (HZI), Campus E8.1, 66123 Saarbrücken, Germany; 4School of Pharmacy, The University of Jordan, Amman 11942, Jordan; 5Department of Pharmaceutical Sciences, School of Pharmacy, University of Puerto Rico—Medical Sciences Campus, San Juan, PR 00921, USA; 6National Centre for Aquatic Animal Health, Cochin University of Science and Technology, Kochi, Kerala 682016, India; 7University of Guam Marine Laboratory, Mangilao, Guam 96923, USA; reyes.andresjoshua@gmail.com (A.J.R.); biggs.js@gmail.com (J.S.B.); 8Skaggs School of Pharmacy and Pharmaceutical Sciences, University of California San Diego, La Jolla, CA 92093, USA

**Keywords:** *Moorena bouillonii*, marine natural products, chemogeography, metabolomics

## Abstract

The tropical marine cyanobacterium *Moorena bouillonii* occupies a large geographic range across the Indian and Western Tropical Pacific Oceans and is a prolific producer of structurally unique and biologically active natural products. An ensemble of computational approaches, including the creation of the ORCA (Objective Relational Comparative Analysis) pipeline for flexible MS^1^ feature detection and multivariate analyses, were used to analyze various *M. bouillonii* samples. The observed chemogeographic patterns suggested the production of regionally specific natural products by *M. bouillonii*. Analyzing the drivers of these chemogeographic patterns allowed for the identification, targeted isolation, and structure elucidation of a regionally specific natural product, doscadenamide A (**1**). Analyses of MS^2^ fragmentation patterns further revealed this natural product to be part of an extensive family of herein annotated, proposed natural structural analogs (doscadenamides B–J, 2–10); the ensemble of structures reflect a combinatorial biosynthesis using nonribosomal peptide synthetase (NRPS) and polyketide synthase (PKS) components. Compound **1** displayed synergistic in vitro cancer cell cytotoxicity when administered with lipopolysaccharide (LPS). These discoveries illustrate the utility in leveraging chemogeographic patterns for prioritizing natural product discovery efforts.

## 1. Introduction

Natural products discovery programs operate with the general goal of detecting and characterizing chemically unique or biologically active substances. A common obstacle in discovery efforts is the rediscovery of known compounds, suggesting a need for tools and techniques that allow researchers to give priority to samples that possess new or otherwise interesting chemical substances. Various strategies have been employed for the dereplication of known chemicals within samples, and for the prioritization of samples based on chemical composition. In this regard, mass spectrometric analyses, usually in combination with liquid chromatography (e.g., LC-MS), have found great utility in natural products research due to the rapidity, small sample size requirements, and high amount of data generated. As a result, a number of approaches and algorithms have been developed to sift through LC-MS data so as to rapidly detect molecules of greater structural novelty and interest. 

PoPCAR (Planes of Principal Component Analysis in R) applies principal component analysis (PCA) to a processed bucket table of sample features, selects outlying samples across different PCA planes, and then leverages the PCA feature loadings to identify the features that make the outlying samples unique [1]. IDBac integrates proteomics and metabolomics data captured via MALDI-TOF MS applied to bacterial colonies on agar plates to classify bacterial strains and distinguish between closely related strains [2]. Global Natural Products Social Molecular Networking (GNPS) is a platform that facilitates the sharing of mass spectral data and provides tools for performing MS^2^-based networking analyses [3]. GNPS continues to expand the repertoire of innovative approaches and techniques that it offers, with recent additions including a pipeline for Feature-Based Molecular Networking (FBMN) [4]. FBMN utilizes a processed bucket table of sample MS^1^ features in conjunction with MS^2^ fragmentation data to produce highly sensitive molecular networks well suited for quantitation and differentiation of isomeric compounds. In addition to these more specific tools, multiple tools are available for the processing and/or statistical analyses of MS-based chemical profile data, including XCMS [5], MZmine [6], and Metaboanalyst [7]. The GNPS classical molecular networking approach [3] is of particular note. While many approaches are sensitive to sample set heterogeneity and rely on specific or consistent sample preparations and data acquisitions in order to provide appropriate results, the classical molecular networking approach is much more flexible, and its outcomes are insulated from imperfect data. This allows classical molecular networking to be used in analyzing datasets that vary across numerous dimensions (instrument type, chromatographic method, sample preparation, etc.), providing many more opportunities for connecting disparate data sources.

The cyanobacterial genus *Moorena* (previously *Lyngbya*, then *Moorea*) is a prolific source of biologically active natural products, with biosynthetic gene clusters accounting for 18% of *Moorena* spp. genomes, on average [8,9,10]. Consistent with this finding, some 70 different isolated and structurally defined compounds have been reported from *M. bouillonii* (Appendix A) [11,12,13,14,15,16,17,18,19,20,21,22,23,24,25,26,27,28,29,30,31,32,33,34,35,36,37,38,39,40,41,42,43,44]. These display a broad structural diversity, and include peptides [41], cyclodepsipeptides [16], macrolides [12] and glycosidic macrolides [35], and lipids [43]. These compounds are also notable for their biological activities, including cytotoxins such as bouillonamide [23], lyngbouilloside [35], multiple lyngbyabellins [12,13], and the exquisitely potent apratoxin A [16]. Other *M. bouillonii* compounds have been reported with cannabimimetic properties, such as columbamides A–C [25] and mooreamide A [43], or as modulators of intracellular calcium mobilization such as alotamide A [14]. *M. bouillonii* has a wide distribution across the tropical Western Pacific and Indian Oceans. However, *M. bouillonii* metabolites have only been described from collections made from a limited number of discrete locations, including Papua New Guinea [14,19,23,25,28,33,34,35,39,41,43], Guam [11,12,15,16,18,20,22,24,29,36,37,42], Palau [11,18,38,40,44], Malaysia [26,27], Palmyra Atoll [13,21], Fiji [31] (The organism in this manuscript is reported as *M. producens*, however the manuscript includes a photo of the organism, which displays a morphology characteristic of shrimp-woven *M. bouillonii*. The 16S rRNA gene-based classification was inconclusive and known compounds previously isolated from *M. bouillonii* were reported.), the Red Sea [17] (The organism in this manuscript is reported as *M. producens*, however the 16S rRNA gene-based classification is inconclusive and known chemistry associated with *M. bouillonii* was reported.), and the islands of southern Japan [30,32]. Collections from these diverse geographical regions differ substantially in their composition of metabolites, suggesting that even though many compounds are already known from *M. bouillonii*, comparing samples of different geographical origin could reveal distributional patterns in chemodiversity that would facilitate the identification of new natural products. 

Much of the previous work connecting natural products chemistry and geography has focused on the latitudinal herbivory-defense hypothesis (LHDH). The LHDH suggests that tropical species display more developed defense phenotypes (including chemical defenses) than temperate species, due to higher levels of biotic stressors [45,46,47]. Studies in both terrestrial organisms [45,46,47,48] and marine organisms [49,50,51] lend support to this hypothesis, but many examples counter to LHDH have also been reported, layering the theory with some degree of controversy while also revealing the complexity of drivers that influence chemical defense [52,53]. Orthogonally, it has become a common strategy to look in underexplored geographical locations in order to find new and unique natural products. This has led natural products discovery efforts to interesting and exotic habitats, including tropical coral reefs [11,12,13,14,15,16,17,18,19,20,21,22,23,24,25,26,27,28,29,30,31,32,33,34,35,36,37,38,39,40,41,42,43,44], hypersaline lakes [54], the Arctic [55] and Antarctic [56], hydrothermal vents [57], and the deep sea [58]. In spite of the acknowledgement that sampling in new geographical locations can allow access to new natural products, there are few examples of systematically applying geographical knowledge in order to inform natural product discovery. However, in one study the crude extracts and fractions from 300 geographically and taxonomically diverse cyanobacterial and algal collections were profiled by LC-MS/MS [59]. Analyses by GNPS classical molecular networking revealed geographic hotspots for chemodiversity, thus allowing for a molecular feature to be prioritized based on its chemogeographical distribution. In this case, it led to the characterization of a new metabolite given the common name yuvalamide A. Another example study focused on cyanobacteria from one specific genus, analyzing 10 samples of *Symploca* spp. collected at different times and in different places. This led to the efficient and targeted discovery of a new sample-specific bioactive natural product, samoamide A [60].

In the present study, we illustrate the value of leveraging geographical patterns in chemodiversity to find previously uncharacterized natural products and apply this strategy to the marine filamentous cyanobacterial species *M. bouillonii*. This is a particularly interesting organism because of its wide geographical range and richness in natural products. To enable analyses and inform current discovery efforts based on legacy data, we were inspired to develop a flexible data pipeline described as the Objective Relational Comparative Analysis (ORCA) of chemical profiles from LC-MS data. Analyses of the LC-MS profiles from geographically disparate chemical extracts of *M. bouillonii*, used in conjunction with GNPS classical molecular networking, allowed for the prioritization of a molecular feature that led to the isolation and characterization of a new compound we called laulauamide (**1**). (The discovery, isolation, and structure elucidation of **1** were presented at the 2017 Annual Meeting of the American Society of Pharmacognosy. The name laulauamide was used for a poster presentation, and the associated abstract can be found under abstract P-219 at the following link [http://asp2017.org/wp-content/uploads/2016/12/ASP20201720Annual20Meeting_web.pdf]). Molecular networks along with detailed MS^2^ fragmentation analyses revealed the presence of an extensive collection of proposed natural analogs. These display diversification through varied combinations of fatty acid side chains at two locations. Assays for biological activity yielded synergistic cytotoxic activity between **1** and lipopolysaccharide (LPS). Late in the performance of this work, a manuscript appeared from another laboratory that reported the isolation and structure elucidation of the main component of this new natural product family, and assigned it the common name “doscadenamide A” [29], a name we retain so as to not create confusion in the literature record.

## 2. Results and Discussion

To allow for the comparison of LC-MS traces of extracts from different collections of *M. bouillonii*, a new pipeline was created called the Objective Relational Comparative Analysis (ORCA) pipeline (https://github.com/c-leber/ORCA) (Figure 1). ORCA is a flexible, modular pipeline that includes capabilities for simple and customizable MS^1^ feature processing. ORCA can also accept any bucket table of samples vs. features as input, allowing for the comparison of data from any source that can be tabulated in such a manner. To accommodate heterogeneous data and to allow for the comparison of diverse datasets, ORCA MS^1^ feature processing starts with an input directory of mzXML files, from which the MS^1^ features are picked and integrated based on the mass-to-charge ratio (*m*/*z*) and a user-selected variant of retention time (rt). Feature picking is parameterized with the user-defined *m*/*z* and rt tolerances, and the peak size and shape parameters. Subsequently, MS^1^ features picked from each sample file are consolidated based on the user-defined *m*/*z* and rt tolerance parameters and are organized into a samples vs. features bucket table containing feature integration values, with options to apply transformations based on the goals of the downstream analyses. 

After processing of the sample MS^1^ features, or the input of an externally generated sample vs. feature bucket table, the vectors of the feature values can then be utilized to initiate a diverse array of analyses, including hierarchical clustering of the samples to gain insights into the relationships between the samples, and univariate feature selection to learn about what specific MS^1^ features are driving the differences between groups of samples. These analyses can then be visualized as dendrograms or heat maps, respectively. ORCA can also be used to generate a list of the most prominent MS^1^ features across samples and to assign putative identifications from a user-supplied spreadsheet, allowing one to efficiently detect expected peaks across many samples, and to quickly determine the mass spectral signature of new potential isolation targets. ORCA was designed for assessing the relationships between heterogeneous samples and generating hypotheses regarding which features are driving these relationships; this makes ORCA a useful framework for not only learning about chemogeographical patterns, but also for comparing chemical profiles across different growth conditions [61], detecting contamination of botanical extracts [62], identifying chemotaxonomic patterns [63], and many other potential uses.

Crude extracts of field-collected samples of *M. bouillonii* from American Samoa, Guam, Kavaratti (Lakshadweep Islands, India), Saipan, and the Paracel Islands (Xisha) in the South China Sea, as well as an in-house culture from Papua New Guinea, were profiled via LC-MS/MS; the resultant chromatograms were used as inputs for MS^1^ feature processing in ORCA. Hierarchical clustering was performed on the MS^1^ features and a dendrogram was produced with a cophenetic correlation coefficient of 0.905, indicating that the displayed structure in the dendrogram is highly correlated to the cosine distances between samples, and thus is representative of the data (Figure 2). The structure in the dendrogram suggests clustering of samples according to geographical region, a phenomenon that has previously been observed across other cyanobacterial samples [59] but has not been specifically reported as a pattern for *M. bouillonii*. It is worth noting that, while samples with shared geographical origin are indeed arranged in clusters together in the dendrogram, the branch points for each geographical cluster are quite large, ranging from 0.4763 cosine distance for the two samples from Guam to 0.7248 cosine distance for where the three samples from Saipan converge. This is likely the result of a combination of the high variability and complexity in the composition of the studied samples, as well as the “curse of dimensionality” that artificially enlarges distance values when large numbers of features are being considered [64]. Classical molecular networking analysis using GNPS provided an orthogonal view, supporting the idea of chemogeographical specificity in these *M. bouillonii* samples, as numerous clusters of location-specific nodes are visible in the resultant network (Figure 3 and Appendix A). Furthermore, hierarchical clustering performed on presence-absence data of the MS^2^ nodes from GNPS, as visualized with a dendrogram (Figure 4), revealed a chemogeographical clustering similar to that produced from the ORCA MS^1^ features (Figure 2). The geographically associated structure in the data, as observable via both ORCA dendrograms, along with the presence of numerous location-specific clusters in the molecular network, led us to generate the hypothesis that the geographically specific distributions of natural products in our samples could be leveraged to identify previously unreported metabolites. The clustering of samples by specific geographical location stimulated further analyses to determine which molecular features were driving the observed geographic clusters, and which peaks were regionally specific. Particular attention was paid to Saipan, as it represented one region from which no new natural products from *M. bouillonii* had been reported in the scientific literature.

One cluster in the GNPS molecular network comprising only nodes originating from the Saipan-collected samples contained a particularly intense node for a feature with *m*/*z* 457.785 (Figure 3). Further investigations in ORCA revealed that this feature was present in high abundance in all samples from Saipan but was undetected or detectable at very low levels in the MS^1^ spectra of samples from all the other studied locales (Appendix A). Additionally, this feature was not dereplicated when queried against all published compounds from *M. bouillonii* at the time (Appendix A) and when searched against the MarinLit database (http://pubs.rsc.org/marinlit/). This intriguing chemogeographic pattern prompted prioritization of this feature for isolation and structure elucidation, ultimately resulting in the characterization of a region-specific metabolite. Based on the specific collection site from which the Saipan samples originated (Laulau Bay, Saipan), we originally termed this metabolite “laulauamide”. A molecular feature with *m*/*z* 721.10 was found to have a very similar geographic distribution. It was detected with high intensity in samples from Saipan, while being undetected or detectable at very low levels in other samples (Appendix A), and thus was another strong driver of the clustering of the Saipan samples. Isolation and analytical characterization revealed that this MS feature was the sodiated adduct of the known compound lyngbyapeptin A [41] (Appendix A).

*M. bouillonii* biomass (1 L sample, 132 g dry biomass yielding 10 g of crude extract) from Saipan’s Laulau Bay (denoted as Saipan_32 in Figure 2 and Figure 4) was thoroughly extracted with 2:1 dichloromethane and methanol, and the resulting crude extract was fractionated over silica using vacuum liquid chromatography. LC-MS/MS analysis of the fractions revealed the MS^1^ feature of interest to be in highest abundance in two relatively polar fractions. Reverse phase HPLC was used to initially isolate 1.5 mg of this compound from the two fractions. 1D and 2D NMR experiments were utilized to establish the planar structure of **1**, with major contributions from the ^1^H-^1^H Correlated Spectroscopy (COSY), ^1^H-^13^C Heteronuclear Single Quantum Coherence (HSQC), ^1^H-^13^C Heteronuclear Multiple Bond Coherence (HMBC), HSQC-Total Correlation Spectroscopy (TOCSY), and long-range ^1^H-^13^C Heteronuclear Single Quantum Multiple Bond Coherence (HSQMBC) data. 

Compound **1** analyzed by high resolution electrospray ionization mass spectrometry (HRESIMS) suggested a molecular formula of C_27_H_40_N_2_O_4_ via observation of the sodiated molecular ion (observed *m*/*z* at 479.2877, calculated 479.2880), indicating 9 degrees of unsaturation. IR absorptions at 1724.31 and 3310.62 cm^−1^ were indicative of carbonyl and alkyne functionalities, respectively. An ultraviolet absorption at 217 nm was suggestive of an α,β-unsaturated carbonyl functionality. By ^13^C NMR analysis, there were three carbonyls at shifts consistent with amides or esters, a highly polarized double bond consistent with one β-oxygenated enone (δ 94.2 and 179.2), and four acetylenic carbons (δ 84.7, 84.6, 68.52, and 68.50), accounting for 8 of the degrees of unsaturation, and thus indicating a monocyclic species.

The ^1^H and ^13^C NMR chemical shifts for the two acetylene groups were highly similar, and by HMBC correlations both had an adjacent methylene group at the same shift (δ 2.18, H_2_-15 and H_2_-24). In one of these two cases, sequential correlations deduced from the ^1^H-^1^H COSY data, and supported by the results of a ^1^H-^13^C HSQC-TOCSY experiment, provided a spin system involving three additional shielded methylene groups at δ 1.44, 1.39, and 1.75 and 1.42 (H_2_-23, H_2_-22, and H_2_-21). The final of these methylene groups was positioned adjacent to a deshielded methine group at δ 3.75 (H_2_-20). By COSY, the methine was determined to be adjacent to a shielded methyl group at δ 1.12 (H_3_-27), and its chemical shift was explained by an HMBC correlation placing it adjacent to an ester or amide carbonyl (δ 176.4, C-19). The spin system of the second acetylene-terminating partial structure was highly similar and partially overlapped but terminated with a more shielded methine proton at δ 2.13 (H-11) with an adjacent methyl group (δ 1.11, H_3_-18) and amide or ester carbonyl (δ 177.0, C-10). Summarizing, two essentially identical 2-methyl-7-octynoic acid structural units were thus defined from highly similar but non-identical data subsets.

The remainder of the molecule was thus composed of C_9_H_14_N_2_O_2_ with 3 degrees of unsaturation resulting from an enone and one ring structure. Two ^1^H NMR singlets (δ 5.04, H-2 and 3.84, H_3_-9) along with a 9-proton connected spin system remained unassigned. The singlet at 3.84 ppm was assignable to a methoxy group at the β-position of the enone by virtue of its relatively deshielded chemical shift and HMBC correlations to the highly deshielded olefinic carbon at δ 179.2 (C-3). The other singlet was thus assigned to the α-position of this enone as it was attached to a shielded olefinic carbon at δ 94.2 (C-2) and showed an HMBC correlation to the carbonyl carbon at δ 170.0 (C-1). As this partial structure accounted for all oxygen atoms in compound **1**, the shielded nature of this carbonyl necessarily required it to be attached to a nitrogen atom, forming an amide. Based on ^1^H and ^13^C NMR chemical shift data (δ 4.64, H-4; δ 59.2, C-4), one terminus of the remaining spin system was assigned to a methine with an attached nitrogen atom. ^1^H-^1^H COSY data, in conjunction with the ^1^H-^13^C HSQC-TOCSY, allowed formulation of four sequential methylene groups. The final methylene was also relatively deshielded (δ 3.22 and 3.13, H_2_-8; δ 39.3, C-8), consistent with its attachment to a nitrogen atom. At this point, all atoms in the molecular formula of compound **1** were accounted for, except for one proton that was attached to a heteroatom by evaluation of the HSQC data (e.g., only 39 protons were found attached to carbon atoms); this was deduced to be an NH as three of the four oxygen atoms were assigned as carbonyls and one as a methylated enol.

HMBC correlations from the two diastereotopic protons at δ 3.13/3.22 ppm (H_2_-8) to the carbonyl at δ 177.0 (C-10) connected these two partial structures. The other 2-methyl-7-octynoic acid was therefore connected to the only remaining heteroatom, the N-atom connected to the δ 170.0 (C-1) carbonyl of the enone functionality. Remaining structural features at this point included the formation of one ring, and placement of a proton on one of the two nitrogen atoms; two possibilities emerged (**1a** and **1b**) (Figure 5). 

Both structural possibilities had features that were attractive and unattractive from a predicted biosynthetic perspective. In **1a**, the fundamental assembly of the PKS derived octynoic acid; its passage to an NRPS to incorporate a lysine residue, followed by a ketide extension, *O*-methylation of the β-enol, and cyclization to a pyrrolidone ring, is well precedented within cyanobacterial natural products [65,66,67]. However, the acylation of a second octynoic acid residue to the lysine side chain nitrogen is an unprecedented event. Alternative structure **1b** has the attractiveness of a regular, predicted PKS(4)-NRPS(glycine)-PKS(3)-NRPS(glycine)-PKS architecture; however, it is quite awkward in requiring several unusual adjustments to the oxidation state of the carbon atoms, and creation of the second 2-methyl-7-octynoic acid residue via a completely different set of biosynthetic steps from the first one.

Modeling of these two alternative cyclization products for ^13^C NMR shifts (see Appendix A for the predicted ^13^C NMR shifts for **1a** and **1b**, respectively) and comparison with those experimentally measured for **1** revealed that both possibilities were reasonably good fits, but the predicted values for 1a tended to be closer to the shifts experimentally derived for compound **1**. For both C-2, the methyl enol carbon (**1a** δ 95.5, **1b** δ 101.4, **1** δ 94.2), and C-3, the deshielded olefinic carbon (**1a** δ 180.7, **1b** δ 171.5, **1** δ 179.2), the fit for alternative **1a** was considerably better. Only at C-6 was the cyclization product proposed in **1b** favored (**1**a δ 24.3, **1b** δ 19.6, **1** δ 20.4). A deeper look into the long-range ^1^H-^13^C HSQMBC data was undertaken. The key proton distinguishing these two possible structures, H-4 at δ 4.64, showed correlations to several resonances, including two of the three carbonyl resonances (δ 170.0, C-1; δ 176.4, C-19) and the β-oxygenated enone (δ 179.2, C-3); these correlations were compatible with structure **1a** (one 2-bond and two 3-bond correlations), while in structure **1b** these correlations would result from one 2-bond, one 4-bond, and one 6-bond ^1^H-^13^C coupling. Furthermore, analysis of the HMBC correlations observed for H-4 and H_2_-8, both from the lysine-derived residue, showed mutual signals with only C-5 at δ 29.0 and C-6 at δ 20.4 that would be consistent with either proposed structure. There were no shared correlations observed between these protons and the equally 3-bond proximal carbonyl in **1b**, nor 3-bond correlations from H-4 to C-8 and H-8 to C-4 that would be reasonably expected to be observed from **1b**, lending further support for **1a** as being the correct structure of **1**.

Compound **1** contains three stereocenters—two associated with the two 2-methyl-7-octynoic acid side chains, and one contained in the central heterocycle. A racemic standard of 2-methyloctanoic acid was derivatized with (*S*)-(+)-2-phenylglycine methyl ester. A chiral standard of (*S*)-2-methyloctanoic was generated via the zirconium-catalyzed asymmetric carbo-alumination (ZACA) reaction [68] of 1-octene to stereoselectively install a methyl group at the C-2 position, followed by an oxidation to 2-methyloctanoic acid and derivatization with (*S*)-(+)-2-phenylglycine methyl ester. Configuration of both 2-methyloctynoic acid moieties of compound **1** was established to be *R* through catalytic hydrogenation, acid hydrolysis, derivatization with (*S*)-(+)-2-phenylglycine methyl ester, and comparison via LC-MS to the generated standards of 2-methyloctanoic acid coupled with the same chiral auxiliary group (Appendix A). Ozonolysis with an oxidative work-up [69], followed by acid hydrolysis, was used to open the heterocyclic ring structure and liberate lysine from compound **1**. The lysine was then derivatized with Marfey’s reagent (l-FDAA) and compared to racemic and l-lysine standards derivatized with the same Marfey’s reagent, indicating an *S* configuration of this residue (Appendix A). The fully elucidated structure of compound **1** was thus determined as in Figure 6.

Low-resolution LC-MS/MS fragmentation data for **1** consistently showed three peaks at *m*/*z* 321, 303, and 168 (Appendix A), which we predicted to represent a side-chain loss, a side-chain loss plus the loss of an amine, and the loss of both side chains plus an amine, respectively. To better understand the fragmentations of compound **1** and use this information for identifying analogs based on repeating the MS^2^ fragmentation patterns, high-resolution MS^2^ fragmentation data were acquired for compound **1**. Numerous fragment peaks were recorded, including peaks observed at *m*/*z* 321.2171, 303.1901, and 168.1016. These values match very well to the calculated monoisotopic masses of the predicted fragment structures shown in Figure 7 (*m*/*z* 321.217, 303.183, 168.102; allowing for hydrogen rearrangements), lending support to our fragmentation hypothesis, and providing a starting point for understanding and proposing the structures of analogs via their fragmentation patterns. 

GNPS classical molecular networking placed compound **1** as a node in a cluster with seven other nodes originating from the Saipan *M. bouillonii* samples (Figure 3), suggesting several naturally occurring analogs were present. ORCA revealed that compound **1** is also present in samples from Guam, though detected with a much lower MS^1^ intensity than in samples from Saipan. This inspired the generation of a more detailed molecular network composed of both crude extracts and fractions from a Saipan sample and a Guam sample (denoted as Saipan_32 and Guam_46 in the above dendrograms), revealing an even larger cluster of potential analogs that contained 33 nodes, including compound **1** (Appendix A). Some nodes in the cluster had very similar masses, which could be the result of an artifact from the particular parameter set selected for the analysis, an artifact of the low resolution MS data analyzed, or be an indicator of isomeric analogs; therefore, further analysis was needed.

Analysis using the GNPS in browser network visualizer suggested that there was a common connection between many of the potential analogs (23 out of 33, including **1**), namely the presence of an MS^2^ fragment peak at *m*/*z* 168 (Appendix A). To facilitate further analysis of MS^2^ spectra and the presence of potential analogs, the ORCA MS^2^ auxiliary pipeline was developed. MS^2^ scans from the Saipan and Guam crude extract and fractions were binned based on precursor mass, and then filtered to only precursor masses with scans that included a *m*/*z* 168 fragment peak. Clustering scans from each relevant precursor mass by cosine distance, paired with manual analysis, allowed the structures of 9 analogs (**2**–**10**) to be proposed (Figure 8; see Appendix A for the proposed structures, consensus spectra, and predicted fragment structures). It must be noted that alternative structural proposals are conceivable for these analogs; however, given the literature precedent for cyanobacteria to produce families of natural products with the same array of variations in desaturation and oxidation as proposed here, e.g., [66,70,71], and the predictable MS^2^ fragmentation spectra observed, these proposals represent the most parsimonious and best supported structural hypotheses. Ambiguities in the remaining related MS^2^ spectra prevent the definitive assignment of carbon chain isomers and positional isomers, and the proposal of additional analogs, but suggest a process of combinatorial biosynthesis in generating this expansive natural product family. While quantities of these minor metabolites in our samples were not sufficient for isolation and further characterization, the total synthesis published alongside the characterization of **1** [29] is very amenable to incorporating alternative side chains, and this could be used for generating these proposed analogs for further study. 

Compound **1** contains unusual structural features that, while having precedent in other cyanobacterial natural products, have not previously been seen together. Terminal alkynes can be found in several other natural products from *Moorena* spp., including jamaicamide B [66], carmabin A [71], and vatiamides A, C, and E [72], but having two is notable. While ribosomally synthesized and post-translationally modified peptides (RiPPs) and NRPS-derived natural products with amino acid subunits are common in cyanobacteria, lysine is not often seen, especially in the natural products of marine cyanobacteria [73,74]. The heterocycle in **1**, composed of an acetate extended amino acid, has been observed in the malyngamides [65], jamaicamides [66], gallinamides [67], and other cyanobacterial natural products, but again, never has it been reported involving a lysine residue. Two curiosities of the biosynthesis of compound **1**, namely the origin of the two 2-methyl octynoic acid residues and the formation of the heterocycle, can be explained by analogy to what is known about the biosynthesis of the jamaicamides [66]. To generate 2-methyl octynoic acid, a fatty-acid desaturase analogous to JamB could act upon an octanoic acid precursor, or a smaller precursor that has been PKS-extended to the appropriate size. The placement of the methyl group in the 2 position suggests incorporation via S-adenosyl methionine (SAM). Formation of the heterocycle likely occurs as the result of an acetate extension of the carboxyl group of lysine, followed by a Claisen-like condensation and cyclization directed by a cyclase analogous to JamQ. As noted above in the discussion of structural possibilities **1**a and **1**b, what is less clear is how 2-methyl octynoic acid is appended to the terminus of the lysine side chain; the peptide bond formed is far from unusual, but its placement suggests enzymatic activity occurring beyond the otherwise linear PKS-NRPS assembly of the molecule.

To further evaluate the relationships of the structural features found together in compound **1** to the known natural product chemical space, we applied a Small Molecule Accurate Recognition Technology (SMART) [75] analysis to search for structurally similar molecules based on HSQC spectra. SMART did not yield any similar compounds with a cosine value higher than 0.84, further revealing the structural uniqueness of compound **1**. We also utilized the structure similarity search function in SciFinder (https://scifinder.cas.org/), which yielded only the sintokamides (Appendix A). The sintokamides share a similar heterocycle and are halogenated natural products from sponges [76]. While not suggested by either structure similarity query, tetramic acids [77] and prostaglandins (PGE_2_, for example) [78] (Appendix A) are two chemical classes that possess some distant level of structural similarity to compound **1**, and this inspired additional bioactivity testing efforts, as described below. 

Structural similarity to tetramic acids inspired in silico antibiotic screening (http://chemprop.csail.mit.edu/) [79]. The known antibacterials C_12_-tetramic acid and C_14_-tetramic acid scored over five times greater than the highest scoring doscadenamide (Appendix A), providing little incentive to further evaluate the doscadenamides for antibiotic activity. 

Compound **1** was assayed for cytotoxicity against human NCI-H460 cells and yielded an IC_50_ > 22 μM, suggesting negligible cytotoxicity. This lack of cytotoxicity, plus some distant structural similarity to prostaglandins, inspired the screening of compound **1** in a Griess assay for anti-inflammation (as well as cytotoxicity) toward murine macrophages RAW264.7 cells at a range of 7–55 μM. Curiously, rather than producing inflammatory or anti-inflammatory effects, compound **1** yielded dose-dependent synergistic cytotoxicity with lipopolysaccharide (LPS). This anomalous result was confirmed through multiple replicates of the assay (Appendix A). 

The doscadenamides were discovered based on global scale patterns in *M. bouillonii* chemical diversity. This illustrates that cyanobacteria harbor intraspecific chemogeographic patterns, and that these patterns can be utilized to direct discovery efforts towards new, regionally specific natural product families. There are many tools available for pursuing chemogeographic and other metabolite patterns in sample sets that can inform discovery efforts, each with their own strengths and limitations. While tools like the ORCA pipeline and GNPS classical molecular networking may be of limited utility in terms of quantitative analyses and effective separation of isomeric features, their flexibility in handling heterogeneous sample sets allows for comparative analyses between samples that could otherwise not be conducted. Furthermore, the intrinsic imperfection of real-world data and the deficiencies inherent to various tools and approaches encourages that an ensemble of tools and approaches be applied. By using ORCA in conjunction with GNPS, we were able to generate convergent results that increased confidence in our conclusions. Converging results from ORCA and GNPS were also helpful in giving confidence to the parameters selected for our analyses; parameter selection is often a challenge when applying computational techniques and requires deep knowledge of the dataset as well as manual validation. The chemogeographic patterns in *M. bouillonii* natural products that are qualitatively presented in this manuscript highlight the opportunity to further explore *M. bouillonii* natural products chemistry and how compounds and compound families are distributed ubiquitously vs. regionally, at different geographical scales. Studying *M. bouillonii* metabolomics in a more controlled, semi-quantitative fashion would allow these patterns to be evaluated more deeply and will be the focus of a future manuscript. 

Doscadenamide A (**1**), when considered in isolation, is a structurally intriguing compound. Being composed of a heterocyclized, acetate-extended amino acid core appended with terminal alkyne containing side chains, it blends structural features common among cyanobacterial natural products with a flair of the unusual: the inclusion of lysine, the dual terminal alkynes, and the acylation of the lysine side chain with one of those terminal alkyne containing side chains. In considering the doscadenamides as a family of cyanobacterial natural products, it is likely they are produced via a seemingly combinatorial addition of different acyl groups to a consistent core structure. From a biosynthetic perspective, this suggests a low level of fidelity in the assembly process. Connecting this family of compounds to the biosynthetic gene cluster responsible for their production would elevate our understanding of how cyanobacteria diversify their natural product arsenals. Since the aforementioned procedure for the total synthesis of compound **1** [29] is amenable to incorporating alternative sidechains, this could be used for generating the nine proposed natural structural analogs reported here (**2**–**10**), as well as for evaluating their activities as quorum sensing modulators [29] and their cytotoxic synergism with LPS.

## 3. Materials and Methods 

### 3.1. General Experimental Procedures

Optical rotation was measured using a JASCO P-2000 polarimeter (Easton, MD, USA), UV/Vis data were obtained using a Beckman DU800 spectrophotometer (Brea, CA, USA), and IR spectra were recorded on a ThermoScientific Nicolet 6700 FT-IR spectrometer (Waltham, MA, USA). NMR experiments were conducted using a JEOL ECZ 500 NMR spectrometer (Akishima, Tokyo, Japan) equipped with a 3 mm inverse probe (H3X), a Bruker AVANCE III 600 MHz NMR with a 1.7 mm dual tune TCI cryoprobe (Billerica, MA, USA), and a Varian VX500 (Palo Alto, CA, USA). NMR data were processed using Mestrenova (Mestrelab, Santiago de Compostela, Spain) and TopSpin (Bruker, Billerica, MA, USA). NMR data were recorded in CDCl_3_ and referenced to the solvent peak (7.260, 77.160). For the low-resolution LC-MS/MS analysis, a ThermoFinnigan Surveyor HPLC System (San Jose, CA, USA) with a Phenomenex Kinetex 5 μm C18 100 × 4.6 mm column (Torrance, CA, USA) coupled to a ThermoFinnigan LCQ Advantage Max Mass Spectrometer (San Jose, CA, USA) in positive ion mode was used. Samples were analyzed using one of two linear gradients from 30% CH_3_CN + 0.1% formic acid to 99% CH_3_CN + 0.1% formic acid in H_2_O + 0.1% formic acid at a flow rate of either 0.6 mL/min or 0.7 mL/min over 32 min or 30 min, respectively. Samples were run at a concentration of 1 mg/mL, with concentrations increased up to 4 mg/mL in situations where the peak intensities were insufficient. For the HiRes-ESI-MS analysis, an Agilent 6230 time-of-flight mass spectrometer (TOFMS) (Santa Clara, CA, USA) with Jet Stream ESI source was used. For HiResMS^2^ fragmentation data, a ThermoScientific Orbitrap XL mass spectrometer (Waltham, MA, USA) with direct infusion of the sample into the Thermo IonMax electrospray interface was used.

Compound isolation was performed using two semi-preparative HPLCs: a Thermo Scientific Dionex UltiMate 3000 HPLC (Waltham, MA, USA) system with automated fraction collector, a Waters HPLC system with 1500 series pumps (Milford, MA, USA), and a 996 photodiode array detector with manual fraction collection. HPLC separation was performed using a Phenomenex Kinetex 5 µm C18 10 × 150 mm column (Torrance, CA, USA) and reverse phase gradients of acetonitrile in H_2_O, with both solvents containing 0.1% (*v*/*v*) formic acid. HPLC grade organic solvents and Millipore Milli-Q system (Burlington, MA, USA) purified water were used.

All reagents, catalysts, and solvents used for the synthetic experiments were purchased in their purest and driest form. All experiments were carried out under an inert atmosphere (Ar) unless otherwise specified. 

National Cancer Institute (NCI) H460 hypotriploid human cells [American Type Culture Collection (ATCC) HTB-177] and RAW 264.7 murine macrophages (ATCC TIB-71) were purchased from the ATCC (Manassas, VA, USA).

### 3.2. Sample Collection

Fifteen benthic filamentous tropical marine cyanobacterial samples were hand-collected via self-contained underwater breathing apparatus (SCUBA) or snorkeling in American Samoa, Guam, Kavaratti (Lakshadweep Islands, India), Papua New Guinea, Saipan, and the Paracel Islands (Xisha) in the South China Sea between the years 2005 and 2018. Samples from all locations besides Papua New Guinea were preserved in 1:1 seawater and either ethyl or isopropyl alcohol, transported back to laboratories, and stored frozen until extraction. The sample from Papua New Guinea was transported back to the laboratory in a culture flask and propagated in seawater (SW) BG-11 media [80]. For additional metadata about these samples, see Appendix A.

### 3.3. Sample Preparation

Cyanobacterial biomass was exhaustively extracted with 2:1 dichloromethane and methanol, concentrated under vacuum, and resuspended in methanol or acetonitrile at a concentration of 1 mg/mL. Samples were prepared for LC-MS/MS analysis via elution through C18 solid phase extraction (SPE) cartridges.

### 3.4. ORCA Pipeline 

Code, data files, and supporting documentation on use and workings of the ORCA pipeline are available at https://github.com/c-leber/ORCA, while the parameter sets used for the various analyses reported in this study are available in Appendix A. ORCA was written in Python [81] and is built off the following Python packages: pandas (0.25.2) [82,83], numpy (1.16.5) [84,85], pyteomics (4.1.2) [86,87], scipy (1.3.1) [88], networkx (2.4) [89], matplotlib (3.0.3) [90], sklearn (0.21.3) [91], and seaborn (0.9.0) [92]. ORCA is available in the form of a Jupyter Notebook [93,94], to facilitate customization and interactive experimentation. Prior to analyses in ORCA, proprietary LC-MS datafiles were converted to mzXML using MSCONVERT (https://bio.tools/msconvert) [95], which is a part of the ProteoWizard Library [96]. MSCONVERT was also used to convert proprietary LC-MS/MS datafiles to mzML for the ORCA MS^2^ Auxiliary pipeline, and to mzXML or mzML for GNPS.

### 3.5. GNPS Classical Molecular Networking 

Molecular networks were created using the online workflow (https://ccms-ucsd.github.io/GNPSDocumentation/) on the GNPS website (http://gnps.ucsd.edu) and were visualized using Cytoscape (3.7.2) (https://cytoscape.org/) [97] and the GNPS in-browser network visualizer. For full accounting of the networking parameter sets, see Appendix A.

### 3.6. Compound Isolation

*M. bouillonii* biomass from Laulau Bay, Saipan, was thoroughly extracted with 2:1 dichloromethane and methanol, yielding 10 g crude extract from 132 g (1 L) biomass. A portion of the crude extract was fractionated over silica with vacuum liquid chromatography and a standardized solvent system protocol (Appendix A). Two relatively polar fractions (fractions F and G) were found to contain the bulk of compound **1**. Reverse phase HPLC was used to isolate 2.6 mg of this compound from these two fractions. A gradient method from 37% to 50% CH_3_CN + 0.1% formic acid in H_2_O + 0.1% formic acid over 60 min at a flow rate of 4 mL/min resulted in the elution of compound **1** starting at a retention time of approximately 38 min.

### 3.7. Planar Structure Characterization

Compound **1**: white solid, ${\left[\alpha \right]}_{D}^{26}$ +17.7 (*c* 0.1, MeOH); UV/Vis (Appendix A); IR (Appendix A). NMR data Appendix A; ^1^H, ^13^C, COSY, HSQC, HMBC, HSQC-TOCSY, and long-range HSQMBC spectra (Appendix A); HR ESIMS (observed *m*/*z* [M + Na]^+^ at 479.2877, C_27_H_40_N_2_O_4_, calculated 479.2880).

### 3.8. Structure Elucidation—Standard Preparation and Derivatization for Configurational Characterization

Methods for ZACA methylalumination-oxidation [68], catalytic hydrogenation [98], ozonolysis [69,99], acid hydrolysis [69,98,99], peptide coupling [98,100], and derivatization with Marfey’s reagent [98,99] were adapted from the literature.

#### 3.8.1. Synthesis of (S)-2-methyloctanoic Acid

To a solution of trimethylaluminum (891 μL, 1.782 mmol) and (+)-(NMI)_2_ZrCl_2_ (23.84 mg, 0.036 mmol) in 1.5 mL of CH_2_Cl_2_ was added a solution of oct-1-ene (100 mg, 0.891 mmol) in 1.5 mL of CH_2_Cl_2_. After stirring overnight at 23 °C, the mixture was treated with a vigorous stream of O_2_ for 1 h at 0 °C and then stirred for 5 h under an atmosphere of O_2_ at room temperature. The reaction mixture was quenched with 1 M HCl, extracted with CH_2_Cl_2_, washed with brine, dried over MgSO_4_ and concentrated. The residue was purified via silica flash column chromatography (20% ethyl acetate/hexanes) to yield (*S*)-2-methyloctan-1-ol (50 mg, 0.347 mmol, 39% yield) as a clear oil. The crude product was used in the next step without further purification.

To a solution of (*S*)-2-methyloctan-1-ol (50 mg, 0.347 mmol) in acetonitrile (1.4 mL) was added *N*-methyl morpholine *N*-oxide (NMO) solution in H_2_O (468 mg, 3.47 mmol) and tetrapropylammonium perruthenate (TPAP) (12.18 mg, 0.035 mmol) sequentially at room temperature and the mixture was stirred for 2 h. The mixture was then concentrated, and the residue passed through a pad of silica gel using hexanes:diethyl ether (3:1) containing 0.1% acetic acid. The eluted solvent was concentrated to yield (*S*)-2-methyloctanoic acid (48 mg, 0.303 mmol, 88% yield). ${\left[\alpha \right]}_{D}^{26}$ +10.0 (c 1.05, MeOH); ^1^H NMR (500 MHz, CDCl_3_) δ 2.46 (ddq, J = 9.7, 6.8, 3.1 Hz, 1H), 1.69 (m, 2H), 1.44 (m, 2H), 1.37–1.24 (m, 6H), 1.19 (m, 3H), 0.89 (m, 3H); ^13^C NMR (126 MHz, CDCl_3_) δ 183.2, 39.6, 33.7, 31.9, 29.4, 27.3, 22.8, 17.1, 14.3. HRESIMS *m*/*z* [M + H]^+^ 159.1394 (calc. for C_9_H_19_O_2_, 159.1385).

#### 3.8.2. Derivatization of 2-methyloctanoic Acid with 2-phenylglycine Methyl Ester

To generate a 1:1 standard mixture of both possible diastereomers of 2-methyloctanoic acid, 6.0 mg (37.9 µmol) of racemic 2-methyloctanoic acid was combined with 1-[bis(dimethylamino)methylene]-1H-1,2,3-triazolo[4,5-b]pyridinium 3-oxid hexafluorophosphate (HATU) (14.4 mg, 37.9 µmol), (*S*)-(+)-2-phenylglycine methyl ester hydrochloride (7.6 mg, 37.9 µmol) and *N*,*N*-diisopropylethylamine (DIPEA) (30 µL) in dimethylformamide (DMF) (300 µL). This was stirred overnight at room temperature and ambient atmosphere. The reaction mixture was then diluted with 1.0 mL of EtOAc, washed with saturated aqueous NH_4_Cl (3 × 1.0 mL), concentrated under vacuum, and prepared for LC-MS analysis. To generate a chiral standard, 1.8 mg (11.4 µmol) of (*S*)-2-methyloctanoic acid was combined with HATU (4.3 mg, 11.4 µmol), (*S*)-(+)-2-phenylglycine methyl ester hydrochloride (2.3 mg, 11.4 µmol) and DIPEA (30 µL) in DMF (300 µL), and stirred overnight at room temperature and ambient atmosphere. The reaction mixture was then diluted with 1.0 mL of EtOAc, washed with saturated aqueous NH_4_Cl (3 × 1.0 mL), concentrated under vacuum, and prepared for LC-MS analysis. The diastereomeric ratio of the chiral standard was 3:1 by area-under-curve analysis.

#### 3.8.3. Derivatization of Lysine with Marfey’s Reagent (FDAA)

To generate a racemic standard, 0.8 mg (4 µmol) of racemic lysine hydrochloride and 0.1 M NaHCO_3_ (200 µL) were added to a solution of l-FDAA (4.4 mg, 16 µmol) in acetone (600 µL). The reaction mixture was sealed in a vial with ambient atmosphere, stirred at 90 °C for 5 min, neutralized with 6 M HCl, concentrated under vacuum, and prepared for LC-MS analysis. To generate a chiral standard, 1.0 mg (6 µmol) of l-lysine monohydrate and 0.1 M NaHCO_3_ (200 µL) were added to a solution of l-FDAA (6.5 mg, 24 µmol) in acetone (600 µL). The reaction mixture was sealed in a vial with ambient atmosphere, stirred at 90 °C for 5 min, neutralized with 6 M HCl, concentrated under vacuum, and prepared for LC-MS analysis.

#### 3.8.4. Derivatization of Compound **1**

Compound **1** (0.5 mg) was combined with 1.0 mg of Pd/C in 1 mL EtOH and stirred under an atmosphere of H_2_ for 8 h. The mixture was filtered through glass wool, rinsed with EtOH (3 × 1.0mL), and concentrated in vacuo.

Hydrogenated compound **1** (0.25 mg) was dissolved in 1 mL CH_2_Cl_2_, into which a stream of ozone gas was bubbled at −78 °C for 25 min. The reaction was concentrated under vacuum, and the residue was treated with 1 mL of 1:2 35% H_2_O_2_:HCOOH at 70 °C and ambient atmosphere for 20 min. The reaction was again concentrated in vacuo, followed by the addition of 1 mL 6 M HCl. The reaction mixture was stirred in a sealed vial with ambient atmosphere overnight at 110 °C, and then concentrated under vacuum. To the residue, a solution of l-FDAA (0.6 mg, 2 µmol) in acetone (200 µL) and 0.1 M NaHCO_3_ (200 µL) were added. The reaction mixture was sealed in a vial with ambient atmosphere, stirred at 90 °C for 5 min, neutralized with 6 M HCl, concentrated in vacuo, and prepared for LC-MS analysis.

Hydrogenated compound **1** (0.25 mg) was dissolved in 1 mL 6 M HCl. This reaction mixture was stirred in a sealed vial with ambient atmosphere overnight at 110 °C, and then concentrated under vacuum. The residue was combined with HATU (0.4 mg, 1 µmol), (*S*)-(+)-2-phenylglycine methyl ester hydrochloride (0.2 mg, 1 µmol), and DIPEA (20 µL) in DMF (200 µL) and stirred for 6 h at room temperature and at ambient atmosphere. The reaction mixture was then diluted with 1.0 mL of EtOAc, washed with saturated aqueous NH_4_Cl (3 × 1.0 mL), concentrated under vacuum, and prepared for LC-MS analysis.

### 3.9. ORCA MS^2^ Auxiliary Pipeline

Code, data files, and supporting documentation on the use and workings of the ORCA MS^2^ Auxiliary pipeline are available at https://github.com/c-leber/ORCA. MS^2^ spectra were binned with the bin_OOM parameter set to 0, and the cutoff parameter for the hierarchical clustering of the MS^2^ scans for a particular precursor mass was set to 0.15. The MS^2^ scans were filtered to only include scans for precursor masses, which contained fragment peaks with *m*/*z* 168, resulting in 78 precursor masses. These precursor masses were individually analyzed via the hierarchical clustering of scan fragmentation patterns followed by the generation of consensus spectra for each cluster. Consensus spectra were manually inspected to detect interpretable fragmentation patterns similar to those of compound **1**.

### 3.10. Bioassays 

Methods for the NCI-H460 cytotoxicity assay [101] and the Griess assay [102,103] were adapted from the literature.

#### 3.10.1. Cytotoxicity Assay of Compound **1** with NCI-H460 Cell Line

NCI-H460 hypotriploid human cells (ATCC HTB-177) were grown in monolayers to near confluence in flasks and then seeded into wells at 3.33 × 10^4^ cells/mL of Roswell Park Memorial Institute (RPMI) medium with standard fetal bovine serum (FBS), 180 μL/well, and incubated for 24 h at 37 °C in 96-well plates. Cells were exposed to compound **1** at ten half log concentrations, the highest being 21.9 μM with 1% dimethyl sulfoxide (DMSO) present, while the lowest was 0.7 nM. Plates were incubated for an additional 48 h and then stained with 3-(4,5-dimethylthiazol-2-yl)-2,5-diphenyltetrazolium bromide (MTT), for 25 min, after which the optical densities were recorded at 630 and 570 nm for each well on a SpectraMax M2 microplate reader with SoftMax^®^ Pro Microplate Data Acquisition and Analysis Software (Molecular Devices, LLC, Version No. M2, Sunnyvale, CA, USA). The test samples were compared with a negative control of 1% DMSO and a positive control of doxorubicin (0.1 μg/mL and 1 μg/mL), both in RPMI medium. Due to the limited availability of compound **1**, fully toxic concentrations were not reached; hence, the resultant dose–response curve was incomplete. Nevertheless, the IC_50_ value for compound **1** is greater than 21.9 μM.

#### 3.10.2. Griess Assay and Cytotoxicity of Compound **1** in RAW 264.7 Cells

RAW 264.7 murine macrophages (ATCC TIB-71) were seeded at 5 × 10^4^ cells in 96-well plates in Dulbecco’s Modified Eagle Medium (DMEM; Gibco, Carlsbad, CA, USA) supplemented with 10% endotoxin-low FBS (HyClone, characterized, Endotoxin: ≤ 25 EU/mL), 190 μL/well, and incubated for 24 h at 37 °C. Compound **1** at concentrations of 55, 28, 14, or 7 μM was applied in triplicate, and after 1 h lipopolysaccharide (LPS from Escherichia coli 026:B6, =10,000 EU/mg, Sigma-Aldrich, Oakville, ON, Canada) was added (0.5 or 1.5 μg/mL) to all wells except those for the LPS-free controls and those for evaluating the pro-inflammatory effects of compound **1**. LPS alone was used as a negative control, whereas the same LPS concentration with 1% DMSO served as the positive control in the Griess assay. After 24 h, Griess reactions (Section 3.10.3) were used to assess NO generation as a proxy for inflammation, and MTT staining (Section 3.10.1) was used to assess cell viability. Doxorubicin at 3.3 μg/mL was used as a positive control for assessing cell viability. Cell survival was calculated as a percentage compared to wells with 1% or 1.5% EtOH and no LPS. A NO concentration standard curve was prepared in Microsoft Excel based on eight serial dilutions of a nitrite standard (0–100 μM) with DMEM. One-way ANOVA and Tukey’s method were used to test for significance in the cell survival results from the assay; high mortality in certain conditions made statistical analyses of the inflammation data inappropriate. Statistical analyses were applied using GraphPad Prism version 8.0.0 for Windows. Batch variability in LPS potency and RAW 264.7 murine macrophage sensitivity, as well as limited availability of compound **1** necessitated using differing reagent concentrations across the biological replicates.

#### 3.10.3. Griess Reaction

Supernatant from each sample well (50 μL) was added to the experimental wells in triplicate. A 1:1 mixture of 1% sulfanilamide solution in 5% phosphoric acid and 0.1% *N*-1-napthylethylenediamine dihydrochloride (100 μL) was added to each well and the plate was incubated in the dark for 20 min. Optical density was measured at 570 nm on a SpectraMax M2 microplate reader. The raw data were exported to a Microsoft Excel work sheet and the concentration of nitrite in the samples was determined by comparison to the standard curve using regression analysis. 

#### 3.10.4. In silico Antibiotic Screening

The simplified molecular-input line-entry system (SMILES) structures of compound **1**, proposed analogs **2**–**10**, C_12_-tertramic acid, and C_14_-tetramic acid were submitted to Chemprop Predict (http://chemprop.csail.mit.edu/predict) [79], using the Antibiotics model checkpoint.

## Figures and Tables

**Figure 1 marinedrugs-18-00515-f001:**
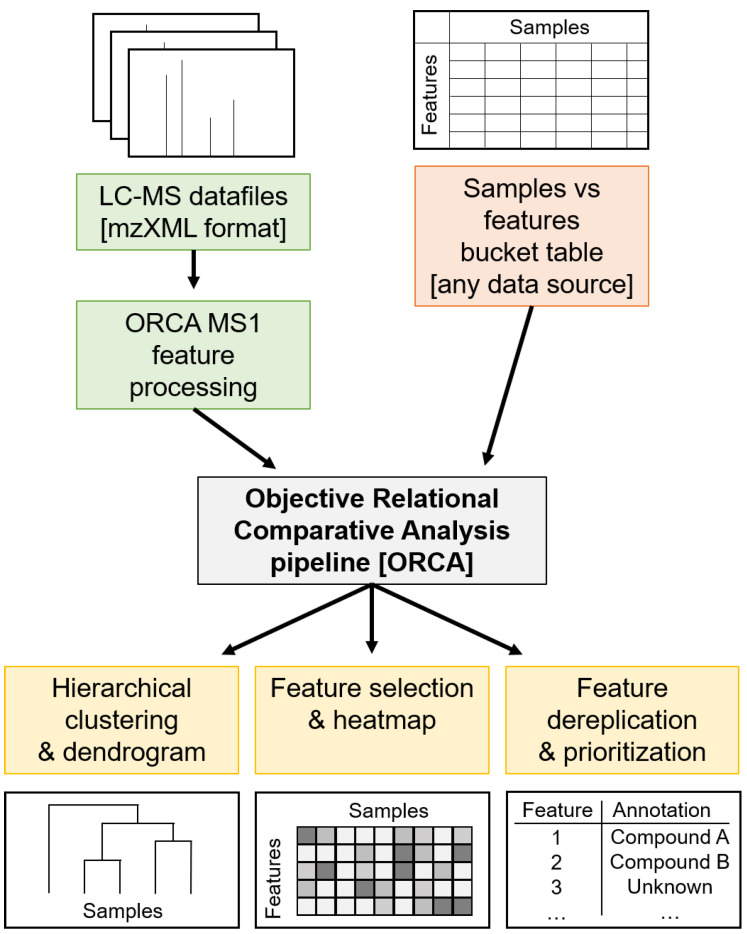
Illustration of the Objective Relational Comparative Analysis (ORCA) pipeline. The pipeline accepts inputs of either LC-MS datafiles in mzXML format, which can then undergo MS^1^ feature processing, or an externally created samples vs. features bucket table coming from any data source. Analyses currently offered as a part of the ORCA pipeline include hierarchical clustering, feature selection, and feature dereplication based on user-provided reference data.

**Figure 2 marinedrugs-18-00515-f002:**
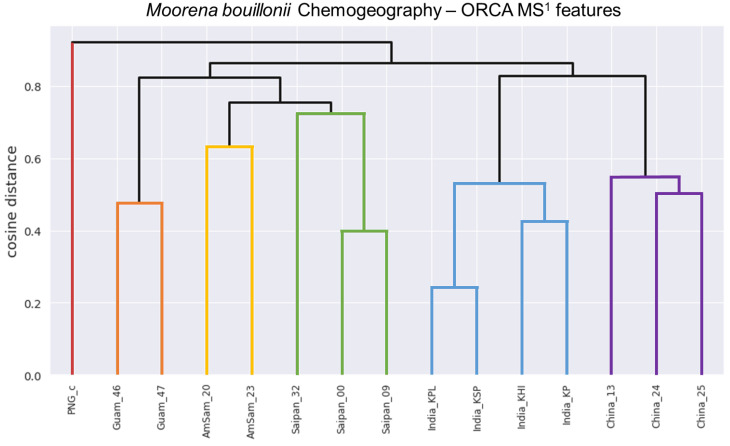
ORCA-generated dendrogram (cophenetic correlation coefficient = 0.905) displaying the results of hierarchical clustering of the MS^1^ features from *M. bouillonii* crude extracts. Samples are labeled with aliases comprising a general collection location concatenated to an abbreviated sample code. The structure in the dendrogram suggests that samples collected from the same geographical area are chemically more similar. Colorized for emphasis. Red: Papua New Guinea; Orange: Guam; Gold: American Samoa; Green: Saipan; Blue: Kavaratti (Lakshadweep Islands, India); Purple: Paracel Islands (Xisha) in the South China Sea.

**Figure 3 marinedrugs-18-00515-f003:**
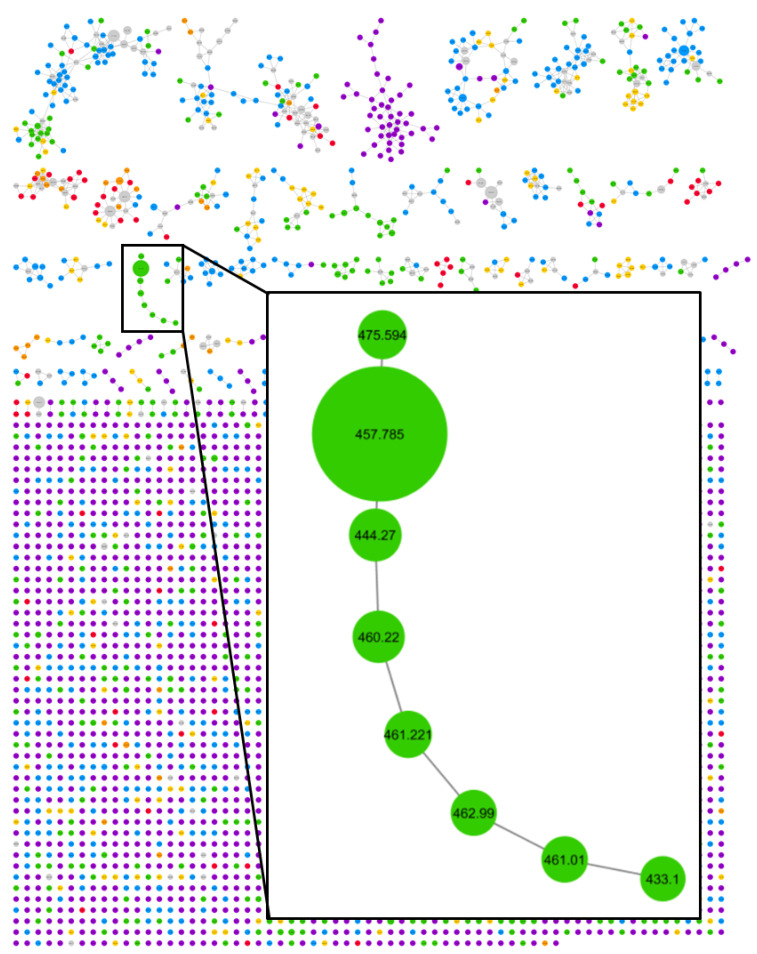
Global Natural Products Social Molecular Networking (GNPS) classical molecular network of fifteen *M. bouillonii* crude extracts with the enlarged inset showing a cluster containing **1** (denoted with precursor mass *m*/*z* 457.785) and seven other nodes representing potential doscadenamide analogs (based on LR-MS/MS data). The green coloring of the nodes indicates that they represent features only detected in samples from Saipan. Nodes are scaled to summed precursor intensity. Grey nodes represent the MS^2^ features that are present in samples from more than one geographical region. Geographical location of samples is colorized as follows: Red: Papua New Guinea; Orange: Guam; Gold: American Samoa; Green: Saipan; Blue: Kavaratti (Lakshadweep Islands, India); Purple: Paracel Islands (Xisha) in the South China Sea.

**Figure 4 marinedrugs-18-00515-f004:**
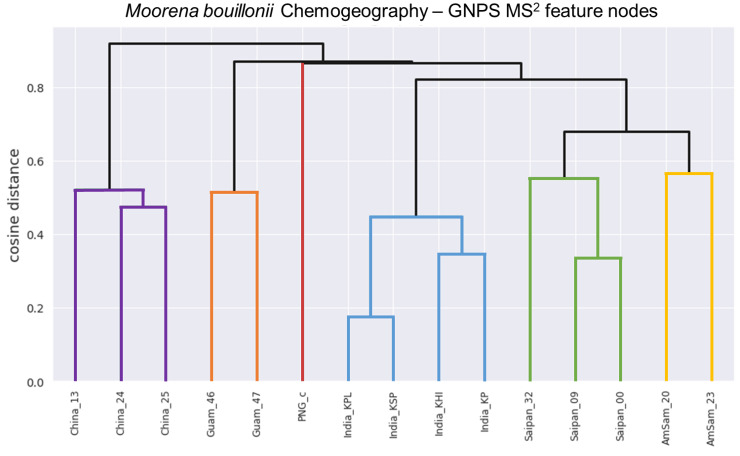
ORCA-generated dendrogram (cophenetic correlation coefficient = 0.960) displaying the results of hierarchical clustering of the *M. bouillonii* crude extracts presence–absence data regarding GNPS nodes. Samples are labeled with aliases comprising a general collection location concatenated to an abbreviated sample code. Similar to Figure 2, the structure in the dendrogram suggests that samples collected from the same geographical area are chemically more similar. Colorized for emphasis. Red: Papua New Guinea; Orange: Guam; Gold: American Samoa; Green: Saipan; Blue: Kavaratti (Lakshadweep Islands, India); Purple: Paracel Islands (Xisha) in the South China Sea.

**Figure 5 marinedrugs-18-00515-f005:**
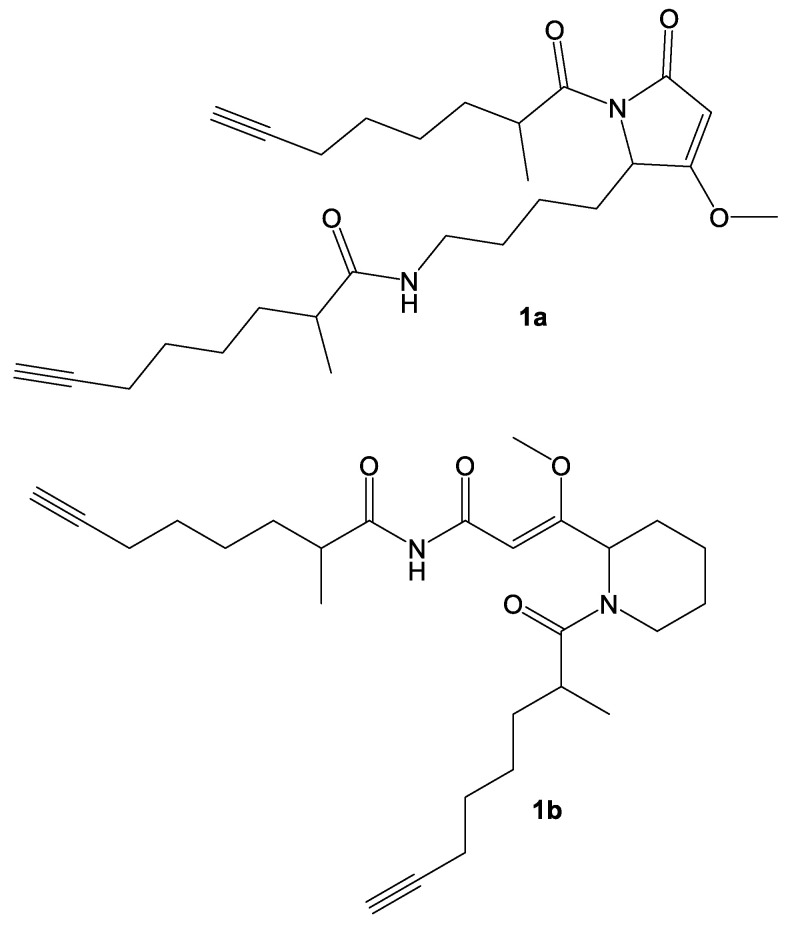
Competing structural hypotheses for the two-dimensional structure of compound **1**.

**Figure 6 marinedrugs-18-00515-f006:**
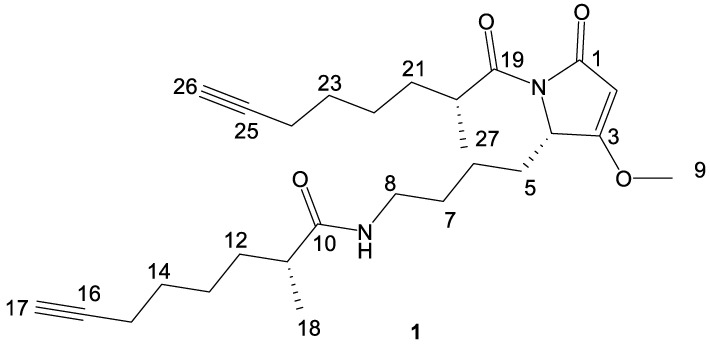
Complete structure of compound **1**.

**Figure 7 marinedrugs-18-00515-f007:**
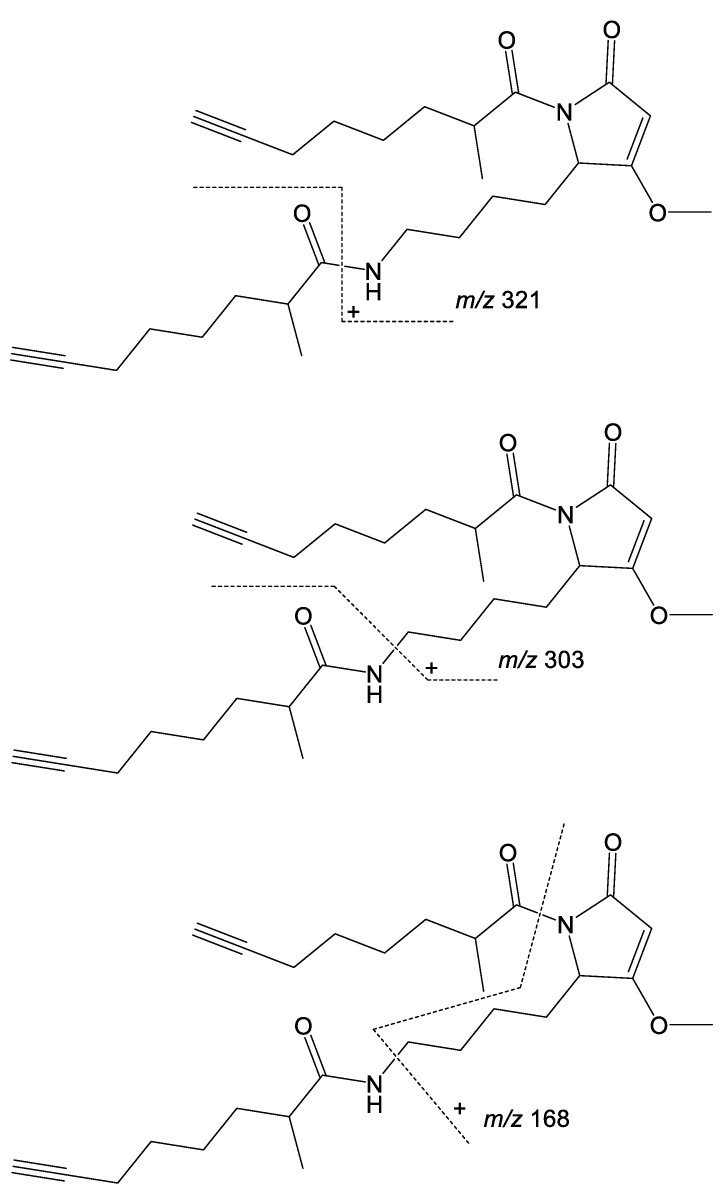
Hypothesized fragment structures of compound **1**.

**Figure 8 marinedrugs-18-00515-f008:**
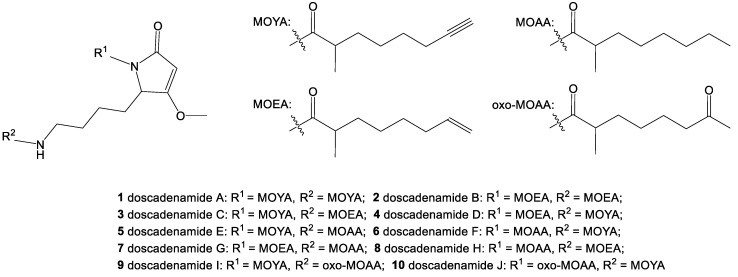
The doscadenamides: compound **1**, along with its analogs whose proposed structures were annotated via informative patterns in the MS^2^ fragmentation data (see Appendix A). Each analog consists of a heterocyclic core with two fatty acid side chains with the following possibilities: MOYA = 2-methyl octynoic acid; MOEA = 2-methyl octenoic acid; MOAA = 2-methyl octanoic acid; oxo-MOAA = 2-methyl 7-oxo octanoic acid.

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
