# Peer review of "Applying a Chemogeographic Strategy for Natural Product Discovery from the Marine Cyanobacterium Moorena bouillonii"

_marinedrugs, 2020, doi:10.3390/md18100515_

Round 1
Reviewer 1 Report
The manuscript describes a novel approach for analysis of natural compounds from cyanobacteria. Hierarchical clustering of data from crude extracts showed that samples of Moorena bouillonii are clustered in the dendrogram according to geographical region of their origin. The authors concluded that doscadenamide A (1) applied together with lipopolysaccharide (LPS).showed synergistic cytotoxicity towards RAW264.7 cells. The manuscript is well prepared and provide a lot of data. Some comments below:
In silico antibiotic screening is not described in methods.
In methods (line 603) 1% EtOH is given, but according to Fig. S27 also 1,5% was used
Missing data in Figs S27-S29-please describe the figures with all details (concentrations of all compounds)-now each graph is prepared in a different way. Some samples are missing in some replicates, e.g. 7μM without LPS in Fig. S28; 55 μM was tested only once (Fig. S29). In Figs S27 and S29 missing concentrations of LPS. In Fig. S29 concentration of 27 μM was used although in methods 28 μM is listed.
Statistical analysis of the results of biological assays should be performed.
Author Response
Comments and Suggestions for Authors
The manuscript describes a novel approach for analysis of natural compounds from cyanobacteria. Hierarchical clustering of data from crude extracts showed that samples of Moorena bouillonii are clustered in the dendrogram according to geographical region of their origin. The authors concluded that doscadenamide A (1) applied together with lipopolysaccharide (LPS).showed synergistic cytotoxicity towards RAW264.7 cells. The manuscript is well prepared and provide a lot of data. Some comments below:
In silico antibiotic screening is not described in methods.
A description of the in silico screening has been added to the methods (lines 699-702).
In methods (line 603) 1% EtOH is given, but according to Fig. S27 also 1,5% was used
Missing data in Figs S27-S29-please describe the figures with all details (concentrations of all compounds)-now each graph is prepared in a different way. Some samples are missing in some replicates, e.g. 7μM without LPS in Fig. S28; 55 μM was tested only once (Fig. S29). In Figs S27 and S29 missing concentrations of LPS. In Fig. S29 concentration of 27 μM was used although in methods 28 μM is listed.
Statistical analysis of the results of biological assays should be performed.
The above listed inconsistencies have all been corrected in the SI, and figures displaying bioassay results have been made more uniform. Batch variability in LPS potency and RAW 264.7 murine macrophage sensitivity, as well as limited availability of compound 1, necessitated using differing reagent concentrations across the biological replicates. We have clearly indicated these differences in the figure captions so as to improve reader comprehension. One-way ANOVA and the Tukey method were applied to determine statistical significance; significance groups are indicated in the caption for each biological replicate, and the methods have been updated to include statistical analyses (lines 684-689).
Reviewer 2 Report
This manuscript describes the structural analysis of natural products from M. bouillonii. Based on the MS and MS2 data of the crude materials which are collected in different countries, the chemogeographic patterns are analyzed and discussed. Doscademamide A which was recently discovered as a new natural product was isolated in this study. In addition, the presence of doscademamide natural congeners is proposed by the MS2 analysis.
From the viewpoint of classical natural product chemistry, I could not support this paper to be published in Marine Drug because of the lack of isolation protocol and analytical data of doscadenamide B~J (NMR data).
The chemogeographic analysis of the cyanobacteria would be a unique approach. It may be useful information for the researchers in marine natural products chemistry.
Round 2
Reviewer 2 Report
The revised version would have worth publishing as a natural product discovery method based on the static analysis.